

# Advanced parallel implementation of the coupled ocean-ice model FEMAO with load balancing

Pavel Perezhogin[1], Ilya Chernov[2], and Nikolay Iakovlev[1]

[1]Marchuk Institute of Numerical Mathematics of the Russian Academy of Sciences, Moscow, Russia
[2]Institute of Applied Math Research, Karelian Research Centre of RAS, Petrozavodsk, Russia

**Correspondence:** Pavel Perezhogin, pperezhogin@gmail.com

**Abstract.** In this paper, we present a parallel version of the finite element model of the Arctic Ocean (FEMAO) configured for the White sea and based on the MPI technology. This model consists of two main parts: an ocean dynamics model and a surface ice dynamics model. These parts are very different in terms of the amount of computations because the complexity of the ocean part depends on the bottom depth, while that of the sea-ice component does not. In the first step, we decided to

locate both submodels on the same CPU cores with the common horizontal partition of the computational domain. The model domain is divided into small blocks, which are distributed over the CPU cores using Hilbert-curve balancing. Partition of the model domain is static (i.e., computed during the initialization stage). There are three baseline options: single block per core, balancing of 2D computations and balancing of 3D computations. After showing parallel acceleration for particular ocean and ice procedures, we construct the common partition, which minimizes joint imbalance in both submodels. Our novelty is using

arrays shared by all blocks that belong to a CPU core instead of allocating separate arrays for each block, as is usually done. Computations on a CPU core are restricted by the masks of not-land grid nodes and block-core correspondence. This approach allows us to implement parallel computations into the model that are as simple as when the usual decomposition to squares is used, though with advances of load balancing. We provide parallel acceleration of up to 996 cores for the model with resolution $500 \times 500 \times 39$ in the ocean component and 43 sea-ice scalars, and we carry out detailed analysis of different partitions on the

model runtime.

## 1   Introduction

The increasing performance and availability of multiprocessor computing devices makes it possible to simulate complex natural systems with high resolution, while taking into account important phenomena and coupling comprehensive models of various subsystems. In particular, more precise, accurate, and full numerical description of processes in seas and oceans have become

possible. There are now models of seas that can simulate currents, dynamics of thermohaline fields, sea ice, pelagic ecology, benthic processes, and so on; see, for example, review Fox-Kemper et al. (2019)

The finite-element model of the Arctic Ocean (FEMAO) Iakovlev (1996, 2012) has been developed since the 1990s and it has been adjusted to the White Sea Chernov (2013), Chernov et al. (2018). The model domain is a part of the cylinder over sphere (i.e., the Cartesian product of a region on the Earth surface to a vertical segment). The coordinates are orthogonal, with





the axes directed to the East, to the South, and downwards. The horizontal grid is structured and rectangular because finite elements are defined on triangles composing rectangles, see Iakovlev (1996). Points that correspond to the land are excluded from the computations using a *mask of "wet" points*. The z-coordinate is used as the vertical axis (i.e., not sigma coordinate and so on). Therefore, for each 2D-grid node, there is the number of actually used vertical layers. In case of significantly variable depth, this "integer depth" may also vary, see figure 1. In contrast, sea ice and sea surface computations are depth-independent. The presence of both 2D and 3D calculations complicates balancing of the full model.

The original code was written in Fortran-90/95 and it did not allow computation in parallel. Our goal is to develop a parallel version of the model based on the MPI technology without the need to make significant changes in the program code (i.e., preserve loops structure, mask of wet points, but benefit from load balancing). Consequently, we developed a library that performs a partition of the 2D computational domain and organizes communication between the CPU cores. In numerical ocean models, the baseline strategy is to decompose domain into squares Madec et al. (2015) or into small blocks, with consequent distribution over the processor cores Dennis (2007, 2003); Chaplygin et al. (2019). Both approaches allow to preserve the original structure of the loops and utilize the direct referencing of neighbouring grid nodes on rectangular grids. Decomposition into small blocks is more attractive from the viewpoint of load balancing, especially for z-coordinate models. Blocks can be distributed using the METIS Karypis (1998) software or simpler algorithms, such as Hilbert curves Dennis (2007). We give preference to partition on blocks, which are distributed using Hilbert curves to make the code library-independent.

Note that some modern ocean models can also benefit from unstructured mesh usage, where there is no need for the mask of wet points; see for example Koldunov et al. (2019). In addition, some ocean models omit masking of wet points, see Madec et al. (2015). This implies increase in the number of computations, but benefits from less control-flow interruptions that give rise to better automatic vectorization of loops. In the following sections we will describe our parallel version implementation *relying* on the use of mask of wet points to make balanced computations and we will also outline its peculiar properties.

## 2 The White Sea

The White Sea is a relatively small (about $500\ \mathrm{km} \times 500\ \mathrm{km}$) and shallow (67 m is the mean depth with a maximal depth of not more than 340 m) semi-closed sea in the Arctic Ocean basin, located in the North-Western part of Russia and included in its territorial waters. Its area is 90000 $\mathrm{km}^2$. The White Sea plays an important role for economy of the neighbouring regions Filatov et al. (2007).

The White sea consists of several parts, including four bays and a narrow shallow strait called Gorlo that separates one part of the sea from the other. The coastline of the sea is quite complex, which means that the rectangle (almost a square) of the Earth's surface that contains the sea has only about one third of the water area.

The White Sea is a convenient model region to test the numerical algorithms, software, and mathematical models that are intended to be used for the Arctic Ocean. First, low spatial step and relatively high maximum velocities demand, due to the Courant stability condition, a rather small temporal step. This makes it difficult to develop efficient algorithms, stable numerical schemes, and ensure performance using the available computers. Second, because this model is less dependent on the initial





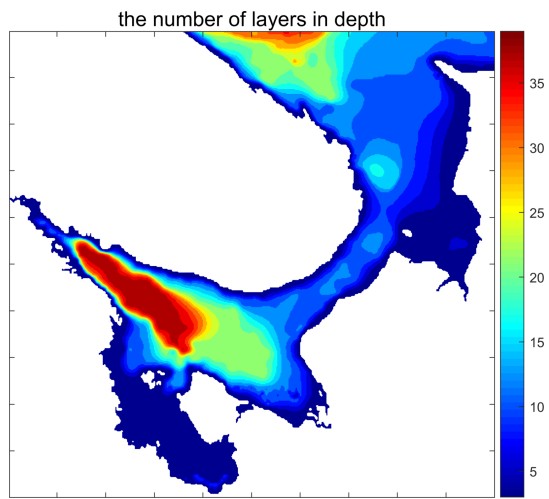

**Figure 1.** The number of depth layers in the White Sea model; the vertical grid has step 5 m up to the 150 m deep and then 10 m up to the 240 m.

distributions, it makes the test simulations easier because the only liquid boundary is needed to set the boundary data. Finally, the White Sea's relatively small inertia enables quite short simulations that are able to reveal any problems and demonstrate

any important features.

## 3 The model and the software

A time step in the FEMAO model consists of several procedures, see algorithm 1. The model uses the physical-process splitting approach, so that geophysical fields are changed by each procedure that simulates one of the geophysical processes.

---

**Algorithm 1** Time step algorithm for FEMAO

---

1: Forcing (i.e., preparation of river runoff, atmospheric data, shortwave radiation, boundary values, etc.);

2: Dynamics of the sea ice, including melting and freezing, interaction of sea-ice floes, and also evaluating the velocity of two-dimensional ice-drift;

3: Sea-ice advection by this drift velocity;

4: Advection of 3D scalars, such as temperature and salinity;

5: Vertical diffusion of the scalars with sources due to heating, ice melting/freezing, and so on;

6: Dynamics of 3D horizontal current velocity;

7: Solving the SLAE for the sea level;

8: Evaluating the vertical velocity.

---

The matrix of the System of Linear Algebraic Equations (SLAE) is sparse and it contains 19 non-zero diagonals that cor-

65 respond to adjacent mesh nodes within a finite element. The matrix does not vary in time and it is precomputed before the





time step loop. The most time-consuming steps for the sequential code version were: 3D advection of scalars, 2D advection of sea-ice fields and solving the SLAE for the sea level.

Sea ice is considered to be an ensemble of multiple floes with some thickness distribution. This distribution is approximated by the discrete one with 15 fixed thickness values (gradations), including zero thickness (open water). Sea ice is described by distribution of its compactness (concentration) for each gradation and ice volume for each gradation (excluding water). In addition, snow-on-ice volume for each gradation is evaluated. Therefore, there are 43 2D sea-ice scalars: ice and snow volume for 14 gradations and sea-ice compactness for 15 ones. Because there are 39 vertical layers in an ocean component, the set of all of the sea-ice data is comparable to a single 3D scalar.

The tested version of the model has a spatial resolution of $0.036°$E, $0.011°$N, which is between $1.0$ and $1.3$ km along parallels and $1.2$ km along a meridian. The number of 2D grid nodes is $500 \times 500$, and only 33% of them are "wet" ($84542$). The time step is 100 s. The vertical step is 5 m up to 150 m deep and then 10 m up to 300 m. In fact, in the bathymetry data (ETOPO Amante and Eakins (2009)) the deepest point of the sea more shallow than it really is, which reduces the actual maximum depth to 240 m.

## 4   Organization of the calculations

Computations in the ocean and sea-ice components are performed using three-dimensional arrays, such as $a(i, j, k)$ or $b(i, j, m)$, where $i, j$ represent the horizontal grid indices, $k$ represents the depth-layer, and $m$ represents the ice gradation. The differential operators are local: only neighbouring grid nodes—that is, $a(i \pm 1, j \pm 1, k)$—are used.

Typical differential operators in the ocean component are organized as shown in algorithm 2, where $N_x = 500, N_y = 500$ and $K(i, j)$ is the number of depth layers. For land points, $K(i, j) = 0$, and $K(i, j) \in [3, 39]$ with approximate mean value 12 for the remaining "wet" points, see figure 1.

---

**Algorithm 2** Typical 3D calculation loop

---

1: **for** $j = 1, N_y$ **do**
2:    **for** $i = 1, N_x$ **do**
3:       **for** $k = 1, K(i, j)$ **do**
4:          $a(i, j, k) = ...$
5:       **end for**
6:    **end for**
7: **end for**

---

Differential operators in the ice component are shown in algorithm 3, where $M = 14$ or 15 is the number of ice gradations and $\text{mask}(i, j)$ is the logical mask of wet points. The percentage of wet points is 33 %.

Note that arrays in Fortran are arranged in the column-major order, so the first index $i$ is linear in memory. The presented arrangement of indices is common for ocean models, see for example NEMO Madec et al. (2015). The loops arrangement



---

**Algorithm 3** Typical 2D calculation loop

---

1: **for** $j = 1, N_y$ **do**

2:      **for** $i = 1, N_x$ **do**

3:          **if** $\text{mask}(i, j)$ **then**

4:             **for** $m = 1, M$ **do**

5:               $b(i, j, m) = ...$

6:             **end for**

7:          **end if**

8:      **end for**

9: **end for**

---

| 1block, 149 cores | hilbert2d, 149 cores | hilbert3d, 149 cores |
|:---:|:---:|:---:|
| 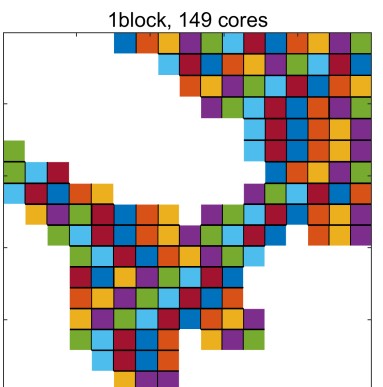 | 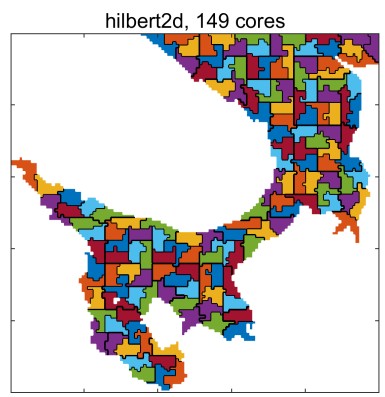 | 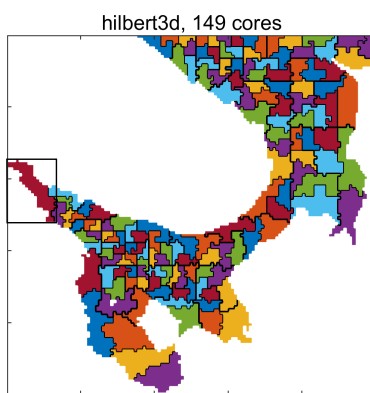 |

**Figure 2.** Three types of partition. Different processor cores are separated by a black line. Hilbert partitions are based on a grid of $n_b \times n_b = 128 \times 128$ blocks. Colours can repeat. Black rectangle corresponds to a "shared" array allocated for blocks belonging to a given CPU core.

is utilized from the original code. Although another arrangement may be more efficient, it does not affect the parallelization approach given later on. In spite of the fact that inner loop does not have stride-1 access, we can speculate that it allows for possible automatic vectorization over $m$ index and corresponds to minimal control flow interruptions due to *false* $\text{mask}(i, j)$ values.

## 5   Modifications of the non-parallel code

Computational domain $[1, N_x] \times [1, N_y]$ is separated into $n_b \times n_b$ blocks. If integer division is impossible, then block sizes are $n_x(i_b, j_b) = N_x \text{ div } n_b$, $n_y(i_b, j_b) = N_y \text{ div } n_b$, where $i_b, j_b \in [1, n_b]$ are horizontal indices of the blocks. The other points are distributed over the first blocks: $n_x(i_b, j_b) += 1$, where $i_b \in [1, N_x \text{ mod } n_b]$, $j_b \in [1, n_b]$ and $n_y(i_b, j_b) += 1$, where $i_b \in [1, n_b]$, $j_b \in [1, N_y \text{ mod } n_b]$. The set of indices corresponding to a block is denoted by $\Omega(i_b, j_b) = [i_s(i_b, j_b), i_e(i_b, j_b)] \times [j_s(i_b, j_b), j_e(i_b, j_b)]$.





To formulate a balancing problem, we must assign weights of computational work to each block and then distribute them among $N_p$ available CPU cores in such a way that all cores have the same amount of work to do, or as close to this as possible, but provided that the "quality" of the partition is kept. Connectivity of subdomains or minimum length of the boundary can be chosen as possible criteria for the quality of a partition. The weight for a block is the sum of weights corresponding to grid points in the range $\Omega(i_b, j_b)$. The following weights are chosen for 2D and 3D computations, respectively:

$$w_{2d}(i,j) = \text{mask}(i,j), \tag{1}$$

$$w_{3d}(i,j) = K(i,j)/\text{mean}(K), \tag{2}$$

where "mean" operation is applied over wet points.

### 5.1 Trivial 1block partition

For a fixed $n_b$, one can find the number of "wet" blocks (i.e., blocks with at least one not-land point). In this partition, the number of cores $N_p$ is equal to the number of wet blocks and each CPU core gets exactly one block, see figure 2. Varying $n_b$, possible values of $N_p$ can be found.

### 5.2 Hilbert curve partition

For $n_b$ being a power of 2, the Hilbert curve connecting all the blocks can be constructed Bader (2012). This gives a one-dimensional set of weights that is balanced using the simplest algorithm. The sum of the blocks' weights on $p$ core is denoted by $W_p$. In spite of the fact that the Hilbert curve possesses the locality property (i.e., close indices on the curve correspond to close indices on the grid), it may not provide a partition into connected subdomains if there are a lot of land blocks. To overcome the problem of possible loss of connectivity, we perform the following optimization procedure, see algorithm 4.

---

**Algorithm 4** Optimization of partition

---

1: remove_not_connected_subdomains();
2: **for** $iter = 1, N_{iter}$ **do**
3: balance_all_ranks();
4: remove_not_connected_subdomains();
5: **end for**

---

Function `remove_not_connected_subdomains()` finds the connected subdomain with the maximum work for each CPU core and sends other blocks to neighbouring cores. Function `balance_all_ranks()` tries to send bordering blocks for each core to neighbouring cores to minimize the maximum work on both cores: $\max(W_p, W_{p'}) \to \min$, where $W_p, W_{p'}$ are for the work on the current CPU core and on a neighbouring core, respectively. The number of iterations is user-defined and we choose $N_{iter} = 15$, which is usually enough to reach convergence. Note that optimization does not guarantee to find a global optimum. The need for partitioning into connected subdomains comes from the intention to increase percentage of the wet points on CPU cores due to the data structure used; see the following section for a definition of the "shared" array.





The described algorithm performs partitioning into connected subdomains with Load Imbalance, which is

$$LI = 100\% \cdot \frac{\max(W_p) - \mathrm{mean}(W_p)}{\mathrm{mean}(W_p)}, \ p \in [1, N_p], \tag{3}$$

not more than 10% in most cases. This is an acceptable accuracy because partitioning itself is not the main objective of the article.

Let us introduce two baseline partitions: hilbert2d (with weights $w_{2d}$) and hilbert3d (with weights $w_{3d}$), see figure 2. As one
can see, hilbert2d divides the computational domain on quasi-uniform subdomains, while hilbert3d locates many CPU cores in high-depth regions and few cores in shallow water. Minimum and maximum number of blocks on a core can be found in tables 2, 3. When one of these partitions is applied to the whole coupled ocean-ice model, it balances one submodel and unbalances another. Table 2 shows that balancing of 2D computations ("LI 2D" → min) leads to imbalance in 3D computations ("LI 3D" $\approx 200\%$) and table 3 shows the opposite behaviour with "LI 2D" $\approx 300\%$. These values are close to the estimates given in
appendix A and defined by the ratio between minimum, maximum and mean integer depth. The presented LI values imply a slowdown of one of the submodels by three to four times because LI increases runtime ($T$) compared to optimal one ($T_{opt}$) in the following way:

$$T = (LI + 1)T_{opt}. \tag{4}$$

A compromise for both submodels can be found by considering a combination of weights:

$$w_{2d3d} = w_{2d} + \gamma_0 w_{3d}, \tag{5}$$

where $\gamma_0 \approx 3$ is a ratio of run times for ocean and ice submodels on one CPU core. A partition of this type is denoted by hilbert2d3d. While this weight is optimal for "overlapping" computations of two code sections with different complexity, it is also the optimal weight for "non-overlapping" code sections (i.e., separated by blocking MPI exchanges). We show this in appendix B with corresponding estimates of LI for 2D (130%) and 3D (34%) computations.

**5.3    Data structure and MPI exchanges**

After partitioning has been performed, we get a set of blocks for each CPU core $p$, $I_p = \{(i_b, j_b)\}$. "Shared" data arrays are allocated for all blocks belonging to a CPU core with the following range of indices (excluding halo):

$$i_s^p = \min(i_s(I_p)), i_e^p = \max(i_e(I_p)), \tag{6}$$
$$j_s^p = \min(j_s(I_p)), j_e^p = \max(j_e(I_p)), \tag{7}$$
$$a(i_s^p : i_e^p, j_s^p : j_e^p, :). \tag{8}$$

The shared array size is shown by rectangle in figure 2 for a particular CPU core. We introduce a mask of grid points belonging to a CPU core ($\mathrm{mask}_p(i,j)$). Correspondence between blocks, shared array and the mask is clarified in figure 3. Introducing this mask does not increase the complexity of the algorithms because the mask of the wet points already exists in the original



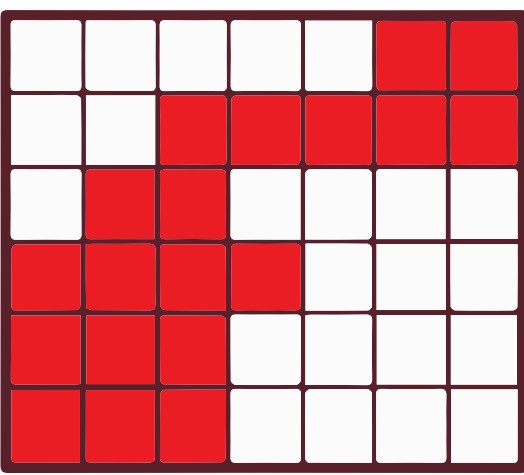

**Figure 3.** Blocks belonging to a CPU core in colour and borders of allocated array by thick line. $\mathrm{mask}_p(i,j) = 1$ in coloured blocks and $0$ elsewhere.

code and it is simply modified. Finally, only minor modifications of the original loops are necessary:

$$1, N_x \to i_s^p, i_e^p, \tag{9}$$

$$1, N_y \to j_s^p, j_e^p, \tag{10}$$

$$K(i,j) \to K(i,j) \cdot \mathrm{mask}_p(i,j), \tag{11}$$

$$\mathrm{mask}(i,j) \to \mathrm{mask}(i,j) \cdot \mathrm{mask}_p(i,j). \tag{12}$$

Usually, see Dennis (2007, 2003), arrays are allocated for each block separately. This has the following advantages:

– More efficient cache usage;

     – If the number of blocks is large enough to get proper balancing, then there is no need for $\mathrm{mask}_p(i,j)$, thus giving advance in vectorization and so on.

It also introduces some drawbacks:

     – Overheads for copying block boundaries (small blocks like $4 \times 4$ are prohibited);

– Many modifications of the original code are necessary, especially in service routines, I/O, and so on.

Consequently, the main strength of our approach is the ability to incorporate balancing while keeping the original program code as simple as for the trivial 1block partition. We expect that a "shared" array may be not optimal for near-land CPU cores with $\sim 20\%$ of wet points because of non-efficient cache usage. An example of such a core is shown by the rectangle in figure 2. An experimental study of runtime dependence on % of wet points will be carried out.





**Table 1.** The model with 1block partition; "$n_b \times n_b$" is the grid of blocks; "LI 2D" and "LI 3D" are load imbalances (3) for weights $w_{2d}$ and $w_{3d}$, respectively. "LI iceadvect" and "LI advect3D" are LIs computed based on the runtime of corresponding functions without exchanges; "days / 24 hours" is the number of computed days for one astronomical day.

| CPU cores | 1 | 32 | 78 | 149 | 306 | 595 | 993 |
|---|---|---|---|---|---|---|---|
| $n_b \times n_b$ | $1^2$ | $7^2$ | $12^2$ | $17^2$ | $26^2$ | $38^2$ | $50^2$ |
| LI 2D, % | 0 | 93 | 62 | 53 | 37 | 28 | 17 |
| LI iceadvect, % | 0 | 80 | 57 | 50 | 40 | 30 | 19 |
| LI 3D, % | 0 | 341 | 317 | 380 | 313 | 278 | 274 |
| LI advect3D, % | 0 | 339 | 324 | 340 | 349 | 290 | 291 |
| days / 24 hours | 8 | 79 | 219 | 360 | 864 | 1763 | 2556 |

Borders of blocks neighbouring with other CPU cores are sent using MPI. The following optimizations are applied to reduce the exchange time:

- All blocks' boundaries adjacent to a given CPU core are copied to a single buffer array, which is sent in one `MPI_Send` call.

- If possible, a diagonal halo exchange is included into cross exchanges with extra width.

- There is an option to send borders of two or more model fields in one `MPI_Send` call. These three bullets reduce latency cost in many cores.

- Borders in the sea component are sent up to $K(i,j)$ depth (i.e., only the actually used layers are transmitted). This reduces bandwidth limitations.

### 5.4 Parallel solver of the SLAE

As we have already mentioned, the time-implicit equation for the free surface is reduced to a SLAE with sparse 19-diagonal matrix. This is solved by a parallel implementation of Bicgstab algorithm preconditioned by block-ILU(0) with overlapping blocks, see Saad (2003). ILU(0) preconditioner preserves the 19-diagonal matrix structure, where matrix blocks are defined for each CPU core and correspond to wet points plus a band of border points of width 2. Because blocks are defined by the partition, the convergence rate depends on the number of CPU cores. Nevertheless, we have found that in the range from 1 to 996 CPU cores, it is sufficient to perform 6 to 10 iterations in order to reach the relative residual $\|Ax - b\|/\|b\| \leq 10^{-6}$.



**Table 2.** Same as table 1, but for hilbert2d partition; min and max operations are applied over CPU cores; column "estimate" shows theoretical LI given in appendix A.

| CPU cores | 1 | 32 | 78 | 149 | 306 | 595 | 993 | estimate |
|---|---|---|---|---|---|---|---|---|
| $n_b \times n_b$ | $1^2$ | $64^2$ | $128^2$ | $128^2$ | $128^2$ | $128^2$ | $128^2$ | |
| min blocks | 1 | 41 | 62 | 34 | 17 | 8 | 5 | |
| max blocks | 1 | 61 | 103 | 54 | 30 | 17 | 11 | |
| min % of wet points | 33 | 20 | 23 | 28 | 27 | 22 | 19 | |
| LI 2D, % | 0 | 9 | 8 | 7 | 4 | 12 | 28 | 0 |
| LI iceadvect, % | 0 | 11 | 19 | 12 | 10 | 19 | 27 | |
| LI 3D, % | 0 | 147 | 195 | 205 | 213 | 222 | 255 | 225 |
| LI advect3D, % | 0 | 145 | 200 | 217 | 242 | 235 | 264 | |
| days / 24 hours | 8 | 129 | 278 | 523 | 890 | 1826 | 2511 | |

**Table 3.** Same as table 2, but for hilbert3d partition.

| CPU cores | 1 | 32 | 78 | 149 | 306 | 595 | 993 | estimate |
|---|---|---|---|---|---|---|---|---|
| $n_b \times n_b$ | $1^2$ | $64^2$ | $64^2$ | $128^2$ | $128^2$ | $128^2$ | $128^2$ | |
| min blocks | 1 | 14 | 5 | 11 | 5 | 2 | 1 | |
| max blocks | 1 | 186 | 80 | 155 | 93 | 55 | 39 | |
| min % of wet points | 33 | 25 | 22 | 22 | 22 | 26 | 21 | |
| LI 2D, % | 0 | 237 | 288 | 268 | 312 | 311 | 305 | 300 |
| LI iceadvect, % | 0 | 238 | 297 | 298 | 373 | 337 | 329 | |
| LI 3D, % | 0 | 5 | 7 | 3 | 12 | 15 | 46 | 0 |
| LI advect3D, % | 0 | 31 | 21 | 18 | 16 | 23 | 48 | |
| days / 24 hours | 8 | 131 | 338 | 691 | 1216 | 2232 | 3130 | |

## 6 Numerical experiments

Our experiments were performed on the cluster of Joint Supercomputer Center of the Russian Academy of Sciences[1]. Each node includes two 16-core processors Intel Xeon E5-2697Av4 (Broadwell). The software code was compiled by the Intel Fortran Compiler ifort 14.0.1 with the optimization option -O2. Simulations were performed for three model days (2592 time steps). During the first day, we call an `MPI_Barrier` function to measure performance of particular procedures with and without exchanges. During the last two days, an `MPI_Barrier` is omitted and overall performance is assessed. The model is launched on 993 CPU cores for 30 days, with subsequent rescaling of the results. The number of cores for tests are guided

---

[1]http://www.jscc.ru/

**Table 4.** Same as table 2, but for hilbert2d3d partition; "estimate" is given in appendix B.

| CPU cores | 1 | 32 | 78 | 149 | 306 | 595 | 993 | estimate |
|---|---|---|---|---|---|---|---|---|
| $n_b \times n_b$ | $1^2$ | $64^2$ | $64^2$ | $128^2$ | $128^2$ | $128^2$ | $128^2$ | |
| min blocks | 1 | 16 | 6 | 14 | 6 | 3 | 2 | |
| max blocks | 1 | 112 | 53 | 104 | 64 | 38 | 23 | |
| min % of wet points | 33 | 22 | 14 | 23 | 23 | 27 | 24 | |
| LI 2D, % | 0 | 95 | 126 | 131 | 130 | 142 | 139 | 130 |
| LI iceadvect, % | 0 | 119 | 138 | 150 | 156 | 150 | 145 | |
| LI 3D, % | 0 | 19 | 27 | 26 | 27 | 41 | 66 | 34 |
| LI advect3D, % | 0 | 24 | 34 | 32 | 29 | 48 | 72 | |
| days / 24 hours | 8 | 180 | 403 | 811 | 1615 | 2718 | 3463 | |

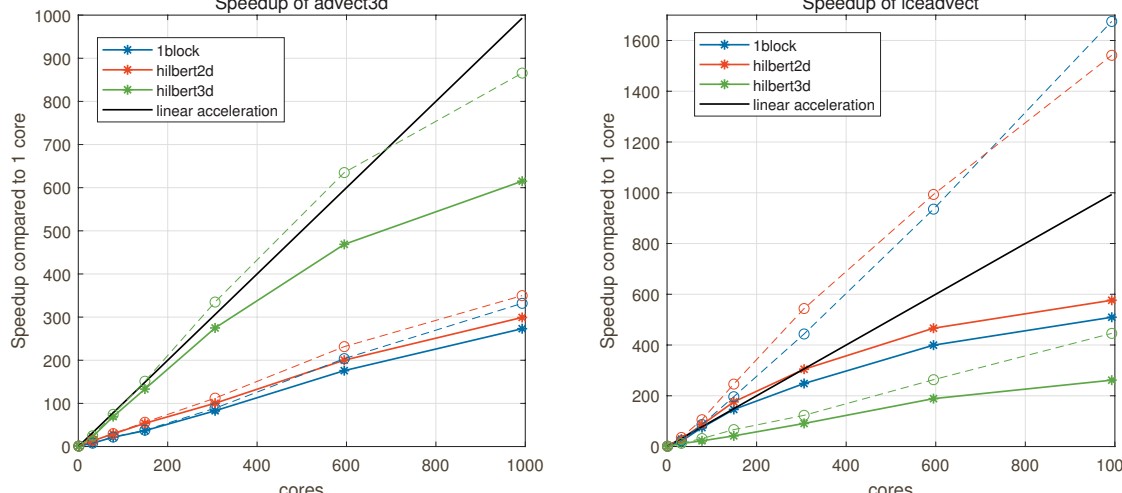

**Figure 4.** Speedup compared to one core for two functions: `advect3D` and `iceadvect`. Solid/dashed lines correspond to measurements with/without MPI-exchanges, respectively. Different partitions are shown in colour (1block, hilbert2d, and hilbert3d).

by the 1block partition method, which is highly restricted in the allowable number of cores. We first show how the most time-consuming functions corresponding to ocean and ice submodels accelerate for three partitions: 1block, hilbert2d and hilbert3d

(see figure 2). We then study overall performance of the model using four partitions, including hilbert2d3d with combined weights (5).

The maximum grid size of blocks for our model is $n_b \times n_b = 128 \times 128$ because the MPI exchange width is limited by the block size, while the SLAE solver requires exchange of width 2. Note that due to the data structure that we used, the



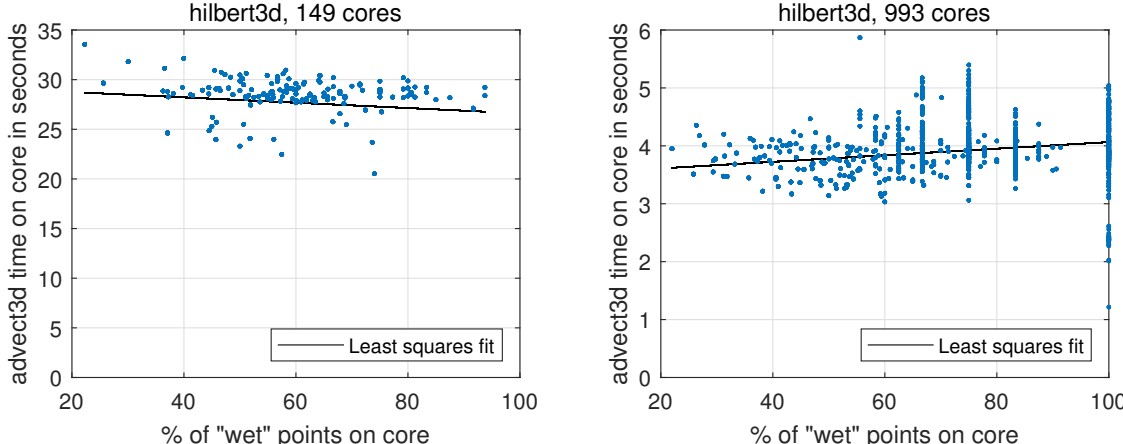

**Figure 5.** Scatter plot: percentage of wet points on a core – `advect3D` runtime (without MPI exchanges, for 1 model day). Each point corresponds to one CPU core. Figures correspond to hilbert3d partition with different numbers of CPU cores.

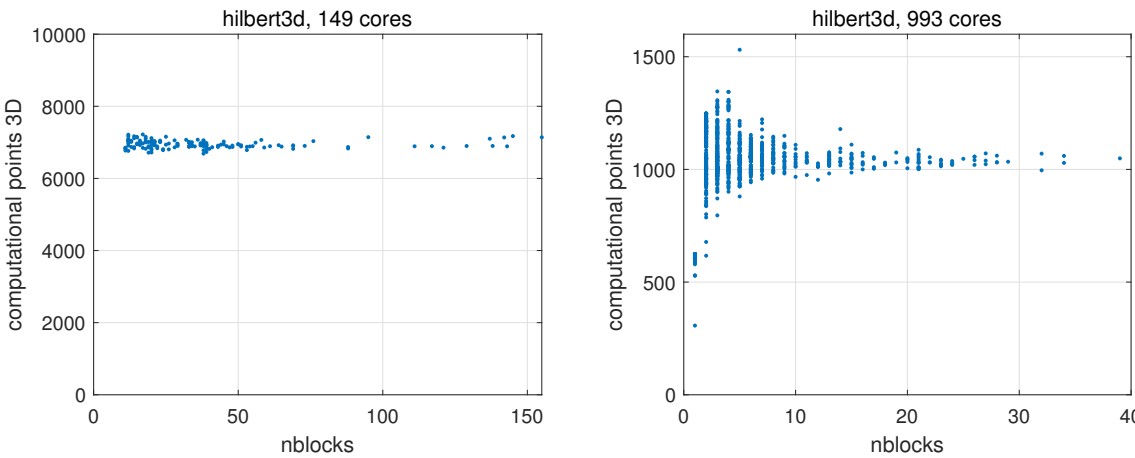

**Figure 6.** Scatter plot: number of blocks on a core – number of computational points for 3D calculations. Each point corresponds to one CPU core. Figures correspond to hilbert3d partition with different numbers of CPU cores.

performance of Hilbert-type partitions is almost insensitive to $n_b$ at moderate number of CPU cores. Nevertheless, $n_b$ may
be tuned by hand to decrease the complexity of the partition optimization procedure or to increase the percentage of the wet points on a core. In runs with many CPU cores, we use the maximum available number of blocks to get better balancing. The parameters that we used in the experiments are given in tables 1, 2, 3, 4.





## 6.1 Speedup of scalar and ice advection

Advection of scalars (`advect3D`, depth-dependent) and ice (`iceadvect`, depth-independent) are the most time-consuming
procedures in ocean and ice submodels, respectively. In the following, we will show that hilbert3d partition is appropriate for
`advect3D` and hilbert2d for `iceadvect`.

Speedup for mentioned procedures is given in figure 4. Dashed lines correspond to measurements of code sections between
MPI exchanges and show how pure computations accelerate. Pure computations in `advect3D` accelerate linearly for hilbert3d
partition, while pure computations in `iceadvect`—superlinearly for the hilbert2d partition. We explain superlinearity by bet-
ter cache usage. The function `advect3D` is only slightly limited by MPI exchanges on 993 cores: its speedup on the partition
hilbert3d falls from 865 to 615 when exchanges are accounted for. Meanwhile, `iceadvect` loses speedup from 1540 to 576
after accounting for exchanges on hilbert2d partition. Both functions have identical number of exchanges, but `advect3D` is
more computationally expensive. Consequently, we explain worse performance of `iceadvect` by lower ratio of number of
operations to the number of points to exchange. Similar bottleneck due to exchanges in 2D dynamics is reported in Koldunov
et al. (2019). The hilbert2d partition has a slight advantage (about 15–20%) over the 1block partition for both functions (see
solid lines). In total, as we expected, the hilbert3d partition is suitable for `advect3D` function, and its acceleration is two
to three times more efficient than when 1block/hilbert2d partitions are used. Also, hilbert2d/1block partitions show two to
four times faster `iceadvect` function compared to hilbert3d partition. The different accelerations are strongly connected to
balancing of computations. To check partition-based ("LI 3D" and "LI 2D") and runtime-based ("LI advect3D" and "LI icead-
vect", correspondingly) Load Imbalance for 3D and 2D computations, see tables 1, 2, 3. Note that theoretical and practical LI
are moderately close to each other, which confirms our choice of weights (1), (2) for these functions. Also note that the data
structure and organization of the calculations are appropriate for load balancing.

Further analysis reveals that the runtime-based LI could be 4–25% more than the partition-based one, see tables 2, 3. This
may be connected to overheads introduced by non-efficient organization of memory. We allocate a shared array for all blocks
belonging to a CPU core and near-land cores may have only 20 % of the wet points (see tables 2, 3), which can lead to an
increase in cache misses. Figure 5 shows a scatter plot for % of the wet points vs `advect3D` runtime without exchanges
(each point corresponds to some CPU core). One can clearly see that on a moderate number of cores (149) the computations
are limited by the core with the smallest % of the wet points, which has the maximal runtime. However, there is no drastic
dependence of runtime on % of wet points. This may mean that the number of operations per one array element in this
model is large enough, thus the data structure plays a moderate role. In particular, the data structure does not limit the model
efficiency on 993 cores (see figure 5), where `advect3D` runtime suffers from imperfect balancing: cores with small number
of blocks usually fall into 100% of wet points and has a wide range of run times, approximately from 1 to 5 seconds. Figure
6 additionally shows that on 993 cores the balancing is limited for processors, where the number of blocks per core is small.
Proper balancing of 3D computations by 2D partitioning implies that the number of blocks per core should be in a wide range,
while the minimum number should not be close to 1 (see the left-hand panel in figure 6). Stagnation of balancing procedure is



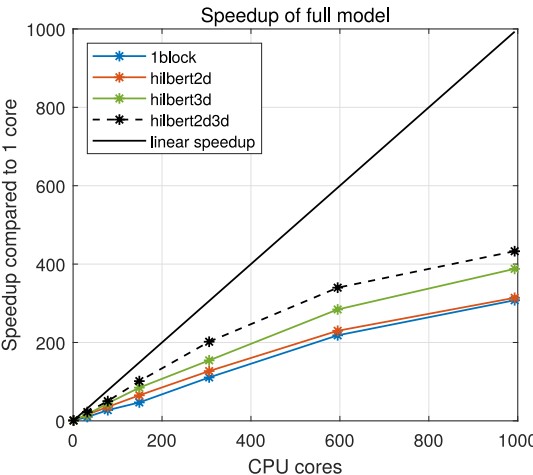

**Figure 7.** Speedup compared to one core for the full model. Different partitions shown in colour (1block, hilbert2d, hilbert3d, hilbert2d3d).

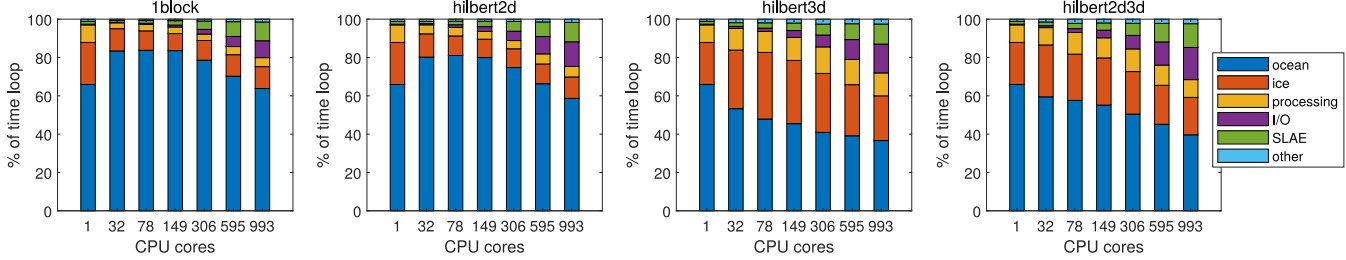

**Figure 8.** Relative contribution of different code sections to runtime; "ocean" – all procedures corresponding to ocean submodel including `advect3D`, "ice" – ice submodel including `iceadvect`, "processing" – computation of statistics, "I/O" – input/output with scatter-gather functions; "SLAE" – matrix inverse and RHS preparation; "other" – simple service procedures.

evident from the fact that we have only $84542$ surface wet points, which corresponds to patches of size $9 \times 9$ (with $12$ vertical levels) on $993$ cores, on average.

## 6.2 Speedup of the full model

The coupled ocean-ice model is launched on the same CPU cores for both submodels with the common horizontal partition.
Input/output functions are sequential and utilize gather-scatter operations, which are given by our library of parallel exchanges. Speedup for the full model compared to one CPU core is given in figure 7. Maximum speedup, approximately $430$, corresponds to the partition with combined weights (hilbert2d3d) on $993$ cores. Compared to the simplest partition (1block), hilbert2d3d model is 115% faster on 149 cores and 40% faster on 993 cores. Partition hilbert2d3d also gives an advantage over partitions balancing purely 2D and 3D computations (hilbert2d, hilbert3d). On 993 CPU cores, the parallel exchanges in this model have
the following contribution to the runtime: 20% for boundary exchanges, 18% for gather-scatter and 6.5% for Allreduce.



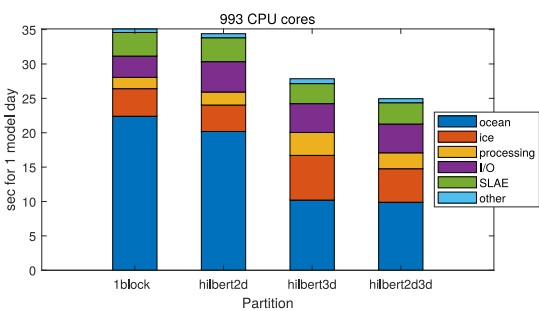

**Figure 9.** Absolute contribution of different code sections to runtime on 993 CPU cores.

The relative contribution of different code sections to runtime is given in figure 8. In the case of perfect scaling of all procedures, the relative contribution must be the same as the number of CPU cores rises. For partitions 1block and hilbert2d, we see a slowdown of the ocean component. Partition hilbert3d suffers from the slowdown of the ice component. Finally, the closest preservation of time distribution is found for hilbert2d3d model. We did not pay much attention to code section 250 "processing" because, although it accelerates, its computational cost could be reduced. Section "I/O" gradually saturates due to gather-scatter operations, which consume 85% of I/O runtime on 993 CPU cores. The new parallel solver ("SLAE") has fast convergence and low computational cost, but suffers from Allreduce operations: in our implementation each iteration demands five `MPI_Allreduce` calls, which account for 60% of "SLAE" code section runtime on 993 cores.

Absolute values of code sections' runtime for 1 model day are shown in figure 9. In comparison to hilbert3d partition, 255 combining of weights (hilbert2d3d) reduces the cost of the ice component, while keeping ocean component almost without changes (see also tables 4, 3 for load imbalance values). In addition, section "processing" reduces its runtime because it contains many not fully optimized service functions that are sensitive to stretching of the horizontal area covered by a CPU core: such stretching is done by the hilbert3d partitioner.

Simulated years per wall-clock day (SYPD) for the best configuration (hilbert2d3d, 993 cores) is $3463/365 \approx 9.5$, see table 260 4. A direct comparison with other coupled ocean-ice models cannot be achieved because our configuration is rare. However, we can rescale the performance (rSYPD) of time step efficiency of the global models in the following way:

$$rSYPD = SYPD \frac{N_{mesh}}{N_{mesh}^{FEMAO}} \frac{\Delta t^{FEMAO}}{\Delta t} \frac{N_p^{FEMAO}}{N_p}, \tag{13}$$

where we take into consideration different numbers of horizontal mesh wet points ($N_{mesh}$), CPU cores ($N_p$) and time step ($\Delta t$), but we neglect different numbers of vertical levels and differences in formulation of the ice dynamics. As follows from table 5, 265 rSYPD is of the order of 10 for all of the ocean-ice models that we have presented, including FEMAO. While this characteristic cannot rate models over their efficiency, we argue that our parallel configuration is comparable to existing parallel ocean-ice models.





**Table 5.** Efficiency of time step loop for FEMAO model compared to global ocean-ice models. Rescaled SYPD (rSYPD, (13)) accounts for difference in the number of horizontal mesh points, CPU cores and time step. Original values are published in Koldunov et al. (2019); Huang et al. (2016); Ward (2016), but we took our values directly from table 3 in Koldunov et al. (2019).

| Model | Mesh points $\cdot 10^6$ | Cores | Time step, s | SYPD | rSYPD |
|-------|---------|-------|--------------|------|-------|
| POP | 5.8 | 16875 | 173 | 10.5 | 24.4 |
| FESOM2/STORM | 5.6 | 13828 | 600 | 15.9 | 12.5 |
| NEMO | 0.9 | 3840 | 1440 | 25.3 | 4.8 |
| MOM5.1 | 0.9 | 3840 | 1800 | 21.6 | 3.3 |
| FESOM2/farc | 0.6 | 2304 | 900 | 56.2 | 19 |
| FEMAO | 0.085 | 993 | 100 | 9.5 | 9.5 |

## 7 Conclusions

In this paper, we present a relatively simple approach to accelerate the FEMAO ocean-ice model based on rectangular structured grid with advances of load balancing. The modifications that had to be introduced into the program code are identical to those that were required by the simplest decomposition on squares. The only demand on the model to be accelerated by this technique is marking computational points by a logical mask. In the first step, we utilize the common partition for ocean and ice submodels. For a relatively "small" model configuration, $500 \times 500$ horizontal points, we reach parallel efficiency of 60 % for particular functions (3D scalar advection using 3D-balancing approach and 2D ice advection using 2D-balancing approach) and 43% for the full model (using combined weight approach) on 993 CPU cores. We show that balancing the 3D computations leads to unbalanced 2D computations, and vice versa. Consequently, further acceleration may be achieved by performing computations of 2D and 3D components on distinct groups of CPU cores with different partitions. Nevertheless, high parallel efficiency of 3D scalar advection itself is a great advance for future applications of the model, especially for the version with a pelagic ecology submodel Chernov et al. (2018), where more than 50 3D scalars (biogeochemical concentrations) are added to the thermohaline fields.

Note that while the parallel approach that we have presented here can be implemented into the model in relatively simple way, the code of the library of parallel exchanges can be rather complex (see Supplements).

*Code availability.* The version of FEMAO model used to carry out simulations reported here can be accessed from https://doi.org/10.5281/zenodo.3977346. The parallel exchanges library with a simple example computing the heat equation is archived on Zenodo https://doi.org/10.5281/zenodo.3873239.





**Appendix A: Estimating the load imbalance for hilbert2d and hilbert3d partitions**

Let us introduce two functions of bathymetry (defined by integer depth $K(i,j)$) with the corresponding values for our model:

$$\rho_{max}(K) = \frac{\max(K)}{\text{mean}(K)} = \frac{39}{12} = 3.25, \tag{A1}$$

$$\rho_{min}(K) = \frac{\text{mean}(K)}{\min(K)} = \frac{12}{3} = 4, \tag{A2}$$

here and below, "mean", "min" and "max" operations correspond only to wet points. These values define how balancing of 2D computations affects 3D computations imbalance, and vice versa. Let $S$ and $V$ be sets of surface and ocean points, correspondingly; $S_p$ and $V_p$ be sets of these points belonging to a CPU core $p$; $|\cdot|$ be the number of points in a set. The number of 3D points can be expressed via 2D ones: $|V| = \sum_{\{i,j\}\in S} K(i,j) = |S| \cdot \text{mean}_{\{i,j\}\in S} K(i,j)$.

When balancing of 2D computations is used (hilbert2d), surface points are distributed among processors in roughly equal
size ($|S_p| = |S|/N_p$). Then, for 3D computations, the ratio of maximum work to mean work among cores is defined as:

$$\frac{W_{max}}{W_{mean}} = \frac{\max_p(|V_p|)}{\text{mean}_p(|V_p|)} = \frac{\max_p(\text{mean}_{\{i,j\}\in S_p} K(i,j))}{\text{mean}_p(\text{mean}_{\{i,j\}\in S_p} K(i,j))} \approx \rho_{max}, \tag{A3}$$

and the corresponding load imbalance is

$$LI = \frac{W_{max} - W_{mean}}{W_{mean}} = \rho_{max} - 1 = 225\%. \tag{A4}$$

When balancing of 3D computations is used (hilbert3d), ocean points are distributed among processors in roughly equal size
($|V_p| = |V|/N_p$). Then, for 2D computations, the ratio of maximum work to mean work is defined as:

$$\frac{W_{max}}{W_{mean}} = \frac{\max_p(|S_p|)}{\text{mean}_p(|S_p|)} = \frac{\max_p(|V_p|/\text{mean}_{\{i,j\}\in S_p} K(i,j))}{\text{mean}_p(|S_p|)} = \frac{|V|/|S|}{\min_p(\text{mean}_{\{i,j\}\in S_p} K(i,j))} \approx \rho_{min}, \tag{A5}$$

and the corresponding load imbalance is

$$LI = \frac{W_{max} - W_{mean}}{W_{mean}} = \rho_{min} - 1 = 300\%. \tag{A6}$$

**Appendix B: Finding the optimal weight for non-overlapping 2D and 3D calculations**

Let $W$ be a full computational work and let it be distributed between 3D ($W^{3d}$) and 2D ($W^{2d}$) computations with ratio $\gamma_0$: $W = W^{2d} + W^{3d} \sim (1+\gamma_0)W^{2d}$. Our goal is to find weight function $w(i,j)$, which corresponds to minimal joint (2D and 3D) Load Imbalance. We use the notation presented in the previous appendix and we define the "number of computational points corresponding to weight": $|V^w| = \sum_{\{i,j\}\in S} w(i,j)$.

Assuming equipartition with respect to this weight ($|V_p^w| = |V^w|/N_p$), we can derive LI for 2D calculations:

$$\frac{W_{max}^{2d}}{W_{mean}^{2d}} = \frac{\max_p |S_p|}{\text{mean}_p |S_p|} \approx \rho_{min}(w), \tag{B1}$$



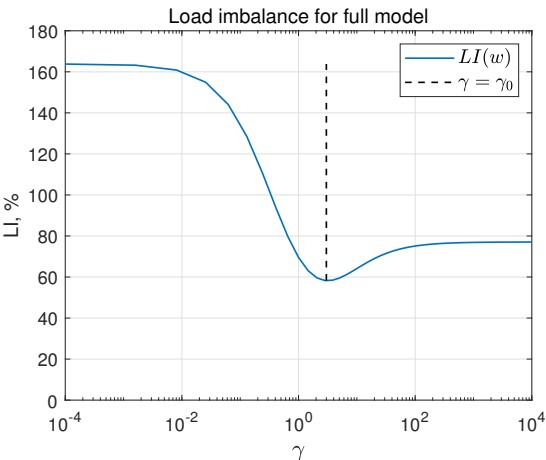

**Figure A1.** Load Imbalance for the full model $L(w)$, as a function of $w(\gamma) = w_{2d} + \gamma w_{3d}$; $\gamma_0 = 3$.

and for 3D calculations:

$$\frac{W_{max}^{3d}}{W_{mean}^{3d}} = \frac{\max_p |V_p|}{\text{mean}_p |V_p|} = \frac{\max_p(|S_p| \cdot \text{mean}_{\{i,j\}\in S_p} K(i,j))}{\text{mean}_p(|S_p| \cdot \text{mean}_{\{i,j\}\in S_p} K(i,j))} = \frac{\max_p \left( \frac{\text{mean}_{\{i,j\}\in S_p}(K(i,j))}{\text{mean}_{\{i,j\}\in S_p}(w(i,j))} \right)}{\text{mean}_p \left( \frac{\text{mean}_{\{i,j\}\in S_p}(K(i,j))}{\text{mean}_{\{i,j\}\in S_p}(w(i,j))} \right)} \approx \rho_{max}(K/w). \quad \text{(B2)}$$

Finally, assuming that 2D and 3D computations are non-overlapping (i.e., the maximum work is under summation), "Load Imbalance" for the full model:

$$LI(w) = \frac{W_{max} - W_{mean}}{W_{mean}} = \frac{\rho_{min}(w) + \gamma_0 \rho_{max}(K/w)}{1 + \gamma_0} - 1. \quad \text{(B3)}$$

For a given bathymetry $K(i,j)$, ratio $\gamma_0 = 3$ and special type of weight function $w(\gamma) = w_{2d} + \gamma w_{3d}$, $LI(w(\gamma))$ can be plotted numerically for different values of $\gamma$, see figure A1. The minimum of this function corresponds to the choice $\gamma = \gamma_0 = 3$, and LI for 2D and 3D computations in this case are $130\%$ and $34\%$, respectively.

*Author contributions.* N. Iakovlev is the developer of the FEMAO model and conceived the research. P. Perezhogin developed the parallel
exchanges library and implemented it in the most computationally expensive parts of the model. I. Chernov completed the implementation
and prepared the test configuration with high resolution. P. Perezhogin performed the experiments and wrote the initial draft of the manuscript.
All of the authors contributed to the final draft of the manuscript.

*Competing interests.* The authors declare that they have no conflict of interest.





*Acknowledgements.* Implementation of the parallel exchanges library into the FEMAO model was carried out with the financial support
of the Russian Foundation for Basic Research (projects 18-05-60184, 19-35-90023). The development of the parallel exchanges library
was carried out with the financial support of Moscow Center for Fundamental and Applied Mathematics (agreement with the Ministry of
Education and Science of the Russian Federation No. 075-15-2019-1624).

We are grateful to G.S. Goyman for helpful remarks on the final draft of the manuscript and assistance in code design. Computing resources
of the Joint Supercomputer Center of the Russian Academy of Sciences (JSCC RAS) were used.



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
