# Peer review of "Advanced parallel implementation of the coupled ocean-ice model FEMAO (version 2.0) with load balancing"

_Geoscientific Model Development, 2020_

## Short Comment (SC1) · 10 Sep 2020

Dear authors,

in my role as Executive editor of GMD, I would like to bring to your attention our Editorial version 1.2:

https://www.geosci-model-dev.net/12/2215/2019/

This highlights some requirements of papers published in GMD, which is also available on the GMD website in the 'Manuscript Types' section:

http://www.geoscientific-model-development.net/submission/manuscript_types.html

In particular, please note that for your paper, the following requirement has not been met in the Discussions paper:

- "The main paper must give the model name and version number (or other unique identifier) in the title."

Please add a version number for FEMAO in the title upon your revised submission to GMD.

Yours,

Astrid Kerkweg
* * *

---

## Referee Comment (RC1) · Nikolay V. Koldunov (Referee) · 1 Oct 2020

The authors present an interesting approach to partitioning the regular grid ocean model, that allows to implement code parallelization in an efficient way by distributing the load between CPU cores in a balanced way. The approach has an advantage of relatively simple implementation, that requires small amount modifications to the original non-parallelised code.

The paper is an interesting contribution to the field and can be published after minor revision.

[Figure]

**Major comments:**

The two step procedure of first dividing the model domain into small blocks and then redistributing those blocks between cores was not really clear to me at first and should be better communicated. It would be helpful for uninitiated readers if you can mention earlier on that the requirement is to preserve the structured nature of the code. So your partitions can't be of arbitrary shape, like in unstructured mesh models, but should be constructed out of small rectangles. I would suggest creating a schematic that shows all the steps of the procedure - splitting into so called blocks, fitting the Hilbert curve, distributing the blocks among CPU cores and finally allocating "shared" arrays. Of course it's not possible to demonstrate with 128x128 blocks you use for a realistic model, but something like a 10x10 schematic representation would do the job.

A bit more details on how the partitioning handled in the model setup would be appreciated. Does the partitioning created by the library and then read by the model? Or it's computed each time. If the latter is the case - do you guarantee that the partitioning will be the same each time the model is run?

**Minor comments:**

Line 23: "adjusted to the White Sea Chernov (2013), Chernov et al. (2018)"

- You forgot parentesis.

Line 28:"(i.e., not sigma coordinate and so on)."

- Just delete it, you don't need this clarification.

Line 28:"In case of significantly variable depth, this "integer depth" may also vary, see figure 1."

- I think I understand what you are trying to say here, the number of levels vary with depth, but it's not clear why depth should be "significantly variable". Please rewrite to make it clearer.

Line 30

- what balancing (I assume you mean balancing of model computation)?

Line 39: "distributed using the METIS Karypis (1998)"

- you need parentheses around citation in this case. Please double check all your citations.

Line 40: "to make the code library-independent"

- my understanding is that you create a separate library for partitioning, so at the end it depends on the library, it's just your library? :)

Line 57:"initial distributions"

- of what? Please be more precise.

Line 59:"demonstrate any important features"

- please rephrase, maybe give some examples.

Line 66

- Please provide details on the type of advection you use.

Lines 68-73.

- Please provide references for sea ice dynamics and thermodynamics.

Line 77:"more shallow than it really is"

- any references to that?

Line 102

- What do you mean by subdomain? Number of blocks that belong to one core? sub-domain in computational domain, like a bay? Please define.

Line 129:"Let us introduce two baseline partitions:"

- change to "We have implemented two baseline partitions"

Fig. 2:"Black rectangle corresponds to a "shared" array"

- change to "Black rectangle on figure for hilbert3d partition corresponds to ..."

Line 151:"The shared array size is shown..."

- Change to "An example of the shared array size..."

Line 189:"Simulations were performed for three model days"

** and then in Line 192

"launched on 993 CPU cores for 30 days"

- Please clarify.

---

## Referee Comment (RC2) · Anonymous Referee #2 · 5 Oct 2020

This paper describes the parallelization of the Finite Element Model of the Arctic Ocean (FEMAO). The focus of the paper is primarily on the technical aspects of domain decomposition and load balancing. While none of the techniques the authors present are new, they have done a couple things a little differently and it's a useful documentation of the model infrastructure so will recommend acceptance with minor revisions.

The main subject of the paper is the MPI implementation and load balancing, but I suspect some aspects of the model are limiting on-node performance. For example, they describe their choice of conditional masking vs multiplicative masking for land points (pg 2 line 42), their non-optimal combination of loop and index ordering (pg

2 L90), and the potential advantages of unstructured meshes (p2, L41). They have included at least some discussion of these in the paper so I'm not suggesting any changes now, but may have some implications on later comments below and would encourage them to explore these as they continue their optimizations in the future.

The focus of the paper is a domain decomposition that subdivides the domain into small blocks that are distributed across MPI ranks. The blocks can be made small and oversubscribed to ranks for better load balancing. The actual partitioning of the mesh uses a space-filling curve approach devised by Dennis. These approaches are typical in the ocean/sea-ice modeling community and appropriate.

A big part of the paper is an exploration of various load balancing techniques. Not surprisingly, they find the load balance is different for the 3-d ocean model vs the quasi-2d sea-ice and find that a weighted blending of the approaches works best. While the current practice in many coupled climate models is to run the two components in different partitions, there is a growing trend to recombine the ice and ocean components, so this weighted load balancing may be of interest to those efforts.

The main difference in the authors' approach is that rather than store these small sub-blocks as individual contiguous haloed arrays as other models have done, they create a larger rectangular domain that geographically covers the area of all the local sub-blocks and creates local index ranges for each of the sub-blocks. This eliminates the need for local halo copies and probably allows them to push toward very tiny sub-blocks (4x4 in many cases presented), but at the expense of contiguous data access and the ability to independently thread over sub-blocks.

Most of the paper is fine, but I remain confused over some of the discussion in section 6.1 and the scatter plots in figures 5,6.

First, the authors show speedups in figure 4 with significantly super-linear speedups in the 2-d case. They attribute this to cache performance without additional evidence (eg from hardware counters or other performance tools). That may be the case, but I think

this super-linearity is large enough to warrant further exploration into the cause.

Second, the computational time as a function of wet points seems a bit counter-intuitive (Fig. 5). The authors have shown percentage of wet points rather than total wet points to emphasize their diagnosis again of memory access. But without also seeing the total number of points (computational load), it's a little hard to get a more complete picture. Again, this effect seems too big to attribute solely to cache effects and it seems like more might be going on here.

Third, the large variation in work load at high core counts (fig 5,6) also seems higher than one might expect. As you get fewer points/blocks per core, there will naturally be a little higher variability, but this seems larger than expected and might point to additional problems.

I suspect some further analysis of the edge cases in the above would help to illustrate what is going on. The issues may go beyond cache performance and may be partly due to the on-node choices mentioned earlier.

A few other minor edits:

The journal editor will probably mention this, but most references should be changed so that the parentheses are around both author and date unless an integral part of sentence. So for example p1L22-23, should have (FEMAO; Iakolev, 1996, 2012) and (Chernov, 2013; Chernov et al. 2018). And so on throughout the manuscript.

Fig 2 has cropped the bottom of figures

P9L170-180; In this bulleted list, move the text "These three bullets..." and "This reduces..." after the bulleted list as "The first three bullets..." and "The final bullet..." Mixing these comments in with the bulleted list was confusing.

---

## Author Comment (AC1) · 12 Nov 2020

Dear editor,

We greatly appreciate your comment. Version (FEMAO 2.0) will be presented in the revised manuscript.

Best regards, On behalf of co-authors, Pavel Perezhogin
* * *

---

## Author Comment (AC2) · 12 Nov 2020

We greatly thank the reviewer for the careful reading of this manuscript and given suggestions.

**1. The two step procedure of first dividing the model domain into small blocks and then redistributing those blocks between cores was not really clear to me at first and should be better communicated. It would be helpful for uninitiated readers if you can mention earlier on that the requirement is to preserve the structured nature of the code. So your partitions can't be of arbitrary shape, like in unstructured mesh models, but should be constructed out of small rectan-**

**gles. I would suggest creating a schematic that shows all the steps of the pro-
cedure - splitting into so called blocks, fitting the Hilbert curve, distributing the
blocks among CPU cores and finally allocating "shared" arrays. Of course it's
not possible to demonstrate with 128x128 blocks you use for a realistic model,
but something like a 10x10 schematic representation would do the job.**

**A bit more details on how the partitioning handled in the model setup would be
appreciated. Does the partitioning created by the library and then read by the
model? Or it's computed each time. If the latter is the case - do you guarantee
that the partitioning will be the same each time the model is run?**

We do not think that there is a need to additionally explain algorithm of distribution of
the blocks over the cores, because it doesn't meet the main objective of the paper, have
been shown many times by Dennis and there is a general-purpose solution (METIS).
"Shared" arrays are clarified in figures 2 and 3. There is no need for another figure.

The introduction is changed: P2 L34 "In numerical ocean model…" is moved to new
paragraph; P2 L39 "We give preference …" is removed; P2 L41 "Note that some
modern…" is moved to previous paragraph.

We add the last paragraph in the introduction: "In sections 2-4 we provide model con-
figuration and organization of the calculations in the non-parallel code on structured
rectangular grid. In section 5 we describe parallelization approach, which preserves
original structure of the loops. Domain decomposition is carried out in two steps: first
the model domain is divided into small blocks and then these blocks are distributed
between CPU cores. For all blocks belonging to a given core a "shared" array is in-
troduced, and mask of computational points restricts calculations. Partition could be
of arbitrary shape, but blocks allow us to reach the following benefits: simple balanc-
ing algorithm (Hilbert curves) can be applied as the number of blocks along a given
direction is chosen to be a power of 2; boundary exchanges can be easily constructed
for arbitrary halo width, but smaller than the block size. In section 6 we report parallel

acceleration on different partitions for particular 2D and 3D subroutines and the whole model."

Section name "Organization of the calculations" is changed to "Organization of the calculations in non-parallel code".

We add the first paragraph to the section "Modifications of the non-parallel code": "In this section we describe the partitioning algorithm of the model domain into subdomains, each corresponding to a CPU core, and subsequent modifications of the single-core calculations, which require only minor changes of algorithms 2 and 3. Grid partition is performed in two steps: model domain is decomposed into small blocks and then these blocks are distributed over CPU cores in such a way that computational load imbalance is minimized. We utilize common grid partition for both sea-ice and ocean submodels, and provide theoretical estimates of the load imbalances resulting from the application of different weight functions in the balancing problem. Partition is calculated during the model initialization stage, as our balancing algorithm (Hilbert curves) is computationally unexpensive. Also, we guarantee that the partition is the same each time the model is run, if parameters of the partitioner were not modified."

2. Minor comments will be taken into account in the revised version of the manuscript.

---

## Author Comment (AC3) · 12 Nov 2020

We are grateful to the referee for the very helpful comments and given suggestions.

1. **The main subject of the paper is the MPI implementation and load balancing, but I suspect some aspects of the model are limiting on-node performance. For example, they describe their choice of conditional masking vs multiplicative masking for land points (pg 2 line 42), their non-optimal combination of loop and index ordering (pg 2 L90), and the potential advantages of unstructured meshes (p2, L41). They have included at least some discussion of these in the paper so I'm not suggesting any changes now, but may have some implications on later comments below and would encourage them to explore these as they continue their optimizations in the future.**

    In this model, boundary conditions are included into matrix elements, which are stored as an array KT(6,13), where the first dimension corresponds to 6 triangles composing Finite Element, and the second dimension corresponds to 13 types of "wet" points: 1 inside the domain and 12 types of boundary points. This approach is similar to multiplicative masking, as B.C.s are applied by the product to KT, and to unstructured mesh models, as matrix elements are precomputed. The difference from the unstructured mesh models is that only unique elements of the matrix are stored and neighboring points are referenced directly. So, the mask of wet points serves only to restrict the number of computations. We thank the referee and will think in the future how to organize calculations more efficiently.

    In our opinion, for the model we have for now, it is not reasonable to change loops' order, as it harms model infrastructure and our parallelization approach, but array indices may be chosen more optimally, setting "depth" index as the first. Nevertheless, this interchange is not crucial for the goals we address in the paper.

2. **First, the authors show speedups in figure 4 with significantly super-linear speedups in the 2-d case. They attribute this to cache performance without additional evidence (eg from hardware counters or other performance tools). That may be the case, but I think this super-linearity is large enough to warrant further exploration into the cause.**

    This super-linear speedup is measured for the code section of approximate length 1000 code lines which consist of 6 loops like in algorithm 3. We stress that these loops have slightly different organization of the calculation and may accelerate in slightly different rates. Some of the loops work with 4D arrays, where the first additional dimension corresponds to "antidiffusive fluxes". Some loops have additional if-conditions, which are needed to perform flux correction in quasi-monotone scheme. Superlinearity occurs at the low-to-middle number of cores, and these cases are usually omitted when scaling up to many cores is shown. Moreover, usually speedups are shown including exchanges, and for this option our speedups are not superlinear.

    Finally, from the practical viewpoint, the presented parallelization approach together with the chosen loops/indices ordering may lead to superlinear acceleration. As an example, consider very simple "heat equation" loop:

    ```
    do j = js, je
        do i = is, ie
            do k = 1,depth(i,j)
                Tn(i,j,k) = 0.25_8 * (T(i+1,j,k) + T(i-1,j,k) + T(i,j+1,k) + T(i,j-1,k))
            end do
        end do
    end do
    ```

It accelerates superlinearly at the low-to-middle number of cores for appropriate weights even when MPI exchanges are taken into account (green line in the right subfigure):

[Figure]

Regardless of the actual reason it happens (decrease in the number of cache misses, non-optimal organization of the calculations, or something else), this "heat equation" loop constitutes what we actually intended to do, and there is nothing to optimize here.

3. **Third, the large variation in work load at high core counts (fig 5,6) also seems higher than one might expect. As you get fewer points/blocks per core, there will naturally be a little higher variability, but this seems larger than expected and might point to additional problems.**

   As the referee pointed out, balancing could be better. In experiment presented in the manuscript, balancing is limited by the outlier point (figs. 5 and 6), corresponding to the CPU core with 5 blocks and maximum load. This outlier point limits LI to 46%. We have checked balancing optimization procedure (algorithm 4) and found that it doesn't guarantee monotone decrease of LI, as subroutine "remove_not_connected_subdomain" can increase LI. After choosing the best iteration, LI was decreased to 29%. As this behavior is crucial only for "hilbert3d 993 cores" experiment, in the revised manuscript we will update it. Additionally, we have tested METIS multilevel k-way contiguous partitioning algorithm and found that it doesn't give better balancing (LI=39%).

4. **Second, the computational time as a function of wet points seems a bit counter-intuitive (Fig. 5). The authors have shown percentage of wet points rather than total wet points to emphasize their diagnosis again of memory access. But without also seeing the total number of points (computational load), it's a little hard to get a more complete picture. Again, this effect seems too big to attribute solely to cache effects and it seems like more might be going on here.**
   Figures 5 and 6 are provided to assess separately data structure efficiency and load balancing efficiency and clearly show limitations of the described model. As the referee pointed out, the lack of point-to-point correspondence between 5 and 6 figures lead to incomplete picture of what is going on. Here we provide scatter plot (6 figure y axis – 5 figure y axis):

[Figure]

[Figure]

Scatterplots are provided with mean values (solid lines) and maximum values (dashed lines). These values completely define Load Imbalance (LI) in partition and advect3d runtime. As follows from the left figure, spread in runtime is more then spread in the number of computational points. This means that computations are limited by the organization of the calculations, but not by the accuracy of the partitioning algorithm. As advect3d is a function with approximate length of 2500 code lines, which consists of 6 loops like in algorithm 2, and each loop has slightly different organization of the calculations, we claim that overestimation of runtime LI only by 15% in comparison to partition LI is a very good result. In the right figure, there is strict correlation between the number of computational points and advect3d runtime, and computations are limited by the balancing procedure. As this figure is more informative than figure 6, in the revised manuscript we will attach the new figure.

5. **Fig 2 has cropped the bottom of figures**

   Fig 2 is shown as we expected. We do not provide axis labels as they correspond to the mesh points, but not to geographical coordinates.

6. Minor edits will be taken into account in the revised version of the manuscript.

---

## Author Response (AR3)

1. **The authors have done a good job addressing most of the issues raised by the referees, and I largely agree with their judgement in choosing to not make some suggested revisions. The only case that I have some disagreement with is their decision to not make any revisions to address the anonymous referee's comments about the superlinearity discussion in Section 6.1. I agree with the referee that this discussion would be strengthened by providing some additional measurements collected with hardware counters to demonstrate the authors' belief that the superlinear speedup is due to improved cache utilization. However, I do not think this is critical, and I respect the authors' decision to not do so. However, I think that the wording here should be amended to make it clear that the authors believe that the superlinear speedup is due to cache effects, but that this has not been verified. Where the authors state "We explain superlinearity by better cache usage", the wording may lead some readers to believe that this explanation has been verified either experimentally or using some cache model. I suggest changing this to something like "We believe the superlinear speedup is due to improved cache usage, but we have not investigated this". It may also be useful to incorporate some of the content they wrote in their justification of their response to the referee on this point, though I leave this to the authors' discretion.**

We greatly appreciate the editor's and anonymous referee's effort to clarify this point. It was helpful to investigate on-node performance using Intel Advisor profiler. Superlinearity occurs in the range of cores 1-149. We have found that for loops in the function iceadvect the time of waiting data from the memory is distributed in the following way for low number of cores:

[Figure]

and for 149 cores:

I.e., memory delay is limited by the DRAM latency and\or bandwidth on low number of cores, and starting from 149 cores most memory accesses correspond to the L1 cache. Analogous behavior for the advect3D function is less crucial, since it is more computationally expansive and rely more on the L1 cache. We think that strict quantitative analysis of data transfer should imply some subsequent reorganization of the calculations, and should be addressed in the future, as the anonymous referee suggests. Thus, it seems enough to give a short qualitative description of this point, and, so, we add the following discussion in Section 6.1.:

"Superlinearity (parallel efficiency is greater than 100% when "doubling" the number of cores) occurs in the range of cores 1-149, and we have investigated the possible reasons for this using the open access Intel Advisor profiler. We have found that on 1 core the most time demanding memory requests are the DRAM data upload, while on 149 cores most memory requests correspond to the L1 cache. Thus, superlinearity can be partially explained by the better cache utilization when the number of cores increases. Note that more local memory access pattern (MAP) can decrease the limitations caused by the memory requests and can be achieved by incorporating stride-1 access for the inner loop indices, but we leave this point for optimizations in the future."

Also, for readability, this discussion is continued with a new paragraph which starts as:

"Speedup including MPI exchanges is shown in figure 4 with solid lines."

**To editor**

Dear editor,

We greatly appreciate your comment. Version (FEMAO 2.0) is presented in the revised manuscript.

Best regards, On behalf of co-authors, Pavel Perezhogin

**To Koldunov**

We greatly thank the reviewer for the careful reading of this manuscript and given suggestions.

1.  **The two step procedure of first dividing the model domain into small blocks and then redistributing those blocks between cores was not really clear to me at first and should be better communicated. It would be helpful for uninitiated readers if you can mention earlier on that the requirement is to preserve the structured nature of the code. So your partitions can't be of arbitrary shape, like in unstructured mesh models, but should be constructed out of small rectangles. I would suggest creating a schematic that shows all the steps of the procedure - splitting into so called blocks, fitting the Hilbert curve, distributing the blocks among CPU cores and finally allocating "shared" arrays. Of course it's not possible to demonstrate with 128x128 blocks you use for a realistic model, but something like a 10x10 schematic representation would do the job.**
    **A bit more details on how the partitioning handled in the model setup would be appreciated. Does the partitioning created by the library and then read by the model? Or it's computed each time. If the latter is the case - do you guarantee that the partitioning will be the same each time the model is run?**

    We do not think that there is a need to additionally explain algorithm of distribution of the blocks over the cores, because it doesn't meet the main objective of the paper, have been shown many times by Dennis and there is a general-purpose solution (METIS). "Shared" arrays are clarified in figures 2 and 3. There is no need for another figure.

    The introduction is changed:
    P2 L34 "In numerical ocean model…" is moved to new paragraph
    P2 L39 "We give preference …" is removed
    P2 L41 "Note that some modern…" is moved to previous paragraph

    We add the last paragraph in the introduction:
    "In sections 2-4 we provide model configuration and organization of the calculations in the non-parallel code on structured rectangular grid. In section 5 we describe parallelization approach, which preserves original structure of the loops. Domain decomposition is carried out in two steps: first the model domain is divided into small blocks and then these blocks are distributed between CPU cores. For all blocks belonging to a given core a "shared" array is introduced, and mask of computational points restricts calculations. Partition could be of arbitrary shape, but blocks allow us to reach the following benefits: simple balancing algorithm (Hilbert curves) can be applied as the number of blocks along a given direction is chosen to be a power of 2; boundary exchanges can be easily constructed for arbitrary halo width, but smaller than the block size. In section 6 we report parallel acceleration on different partitions for particular 2D and 3D subroutines and the whole model."

Section name "Organization of the calculations" is changed to "Organization of the calculations in non-parallel code".

We add the first paragraph to the section "Modifications of the non-parallel code":
"In this section we describe the partitioning algorithm of the model domain into subdomains, each corresponding to a CPU core, and subsequent modifications of the single-core calculations, which require only minor changes of algorithms 2 and 3. Grid partition is performed in two steps: model domain is decomposed into small blocks and then these blocks are distributed over CPU cores in such a way that computational load imbalance is minimized. We utilize common grid partition for both sea-ice and ocean submodels, and provide theoretical estimates of the load imbalances resulting from the application of different weight functions in the balancing problem. Partition is calculated during the model initialization stage, as our balancing algorithm (Hilbert curves) is computationally unexpensive. Also, we guarantee that the partition is the same each time the model is run, if parameters of the partitioner were not modified. "

2. **Minor comments.**
   1) Line 23: "adjusted to the White Sea Chernov (2013), Chernov et al. (2018)" - You forgot parentesis.
      **Response**: We add parenthesis.
   2) Line 28:"(i.e., not sigma coordinate and so on)." - Just delete it, you don't need this clarification.
      **Response**: Deleted.
   3) Line 28:"In case of significantly variable depth, this "integer depth" may also vary, see figure 1." - I think I understand what you are trying to say here, the number of levels vary with depth, but it's not clear why depth should be "significantly variable". Please rewrite to make it clearer.
      **Response**: Rewritten: "In case of significantly variable depth, the number of levels also varies, see figure 1."
   4) Line 30 - what balancing (I assume you mean balancing of model computation)?
      **Response:** Rewritten: "The presence of both 2D and 3D calculations complicates *balancing of the computations* for the full model."
   5) Line 39: "distributed using the METIS Karypis (1998)" - you need parentheses around citation in this case. Please double check all your citations.
      **Response:** parentheses are added. We have checked all citations.
   6) Line 40: "to make the code library-independent" - my understanding is that you create a separate library for partitioning, so at the end it depends on the library, it's just your library? :)
      **Response:** This sentence is removed. Instead, we add the final paragraph in the Introduction with more accurate description of our approach.
   7) Line 57:"initial distributions" - of what? Please be more precise.
      **Response:** Sentence is rewritten: "Second, because this model is less dependent on the initial data, it makes the test simulations easier because the only liquid boundary is needed to set the initial-boundary data."
   8) Line 59:"demonstrate any important features" - please rephrase, maybe give some examples.
      **Response:** Rewritten: "Finally, the White Sea's relatively small inertia enables to check correctness of the code by rather short simulations, which are able to demonstrate important features of the currents."
   9) Line 66 - Please provide details on the type of advection you use.

**Response:** Line 73 of the revised manuscript: "The simple Characteristic-Galerkin Scheme (Zienkiewicz and Taylor,2000) is used for the 3D and 2D advection terms."

10) Lines 68-73. - Please provide references for sea ice dynamics and thermodynamics.

**Response:** Line 75 of the revised manuscript. Paragraph is rewritten:

"The local 1D sea ice thermodynamics is based on the 0-layer model (Semtner, 1976; Parkinson and Washington, 1979) with some modifications in lateral melting and surface albedo (Yakovlev, 2009). There are 14 categories of ice thickness (gradations), the mechanical redistribution and the ice strength are identical to the CICE (Hunke et al., 2013). The elastic-viscous plastic scheme (EVP; Danilov et al., 2015) with modification for the relaxation time scales (Wang et al., 2016) is used for the sea ice dynamics (see also the Appendix 3 in Koldunov et al., 2019b). Sea ice is described by distribution of 80 its compactness (concentration) and ice volume for each gradation. In addition, snow-on-ice volume for each gradation is evaluated. Therefore, there are 43 2D sea-ice scalars: ice and snow volume for 14 gradations and sea-ice compactness for 15 ones (including water). Because there are 39 vertical layers in an ocean component, the set of all of the sea-ice data is comparable to a single 3D scalar."

11) Line 77:"more shallow than it really is" - any references to that?

**Response:** Reference is provided: "Comparison of available bathymetry data for the White Sea is given in Chernov and Tolstikov (2020) in table 1."

12) Line 102 - What do you mean by subdomain? Number of blocks that belong to one core? subdomain in computational domain, like a bay? Please define.

**Response:** Rewritten: "Connectivity of subdomains (by subdomain we refer to a set of blocks belonging to a CPU core) or minimum length of the boundary can be chosen as possible criteria for the quality of a partition."

13) Line 129:"Let us introduce two baseline partitions:" - change to "We have implemented two baseline partitions"

**Response:** changed.

14) Fig. 2:"Black rectangle corresponds to a "shared" array" - change to "Black rectangle on figure for hilbert3d partition corresponds to ..."

**Response:** Rewritten: "Black rectangle in figure for hilbert3d partition corresponds to …"

15) Line 151:"The shared array size is shown..." - Change to "An example of the shared array size..."

**Response:** changed**.**

16) Line 189:"Simulations were performed for three model days" ** and then in Line 192 "launched on 993 CPU cores for 30 days" - Please clarify.

**Response:** These sentences are rewritten:

"Low-core simulations were performed for three model days (2592 time steps). The model on 993 CPU cores is launched for 30 days, with subsequent rescaling of the results."

**To anonymous referee.**

We are grateful to the referee for the very helpful comments and given suggestions.

1. **The main subject of the paper is the MPI implementation and load balancing, but I suspect some aspects of the model are limiting on-node performance. For example, they describe their choice of conditional masking vs multiplicative masking for land points (pg 2 line 42), their non-optimal combination of loop and index ordering (pg 2 L90), and the potential advantages of unstructured meshes (p2, L41). They have included at least some discussion of these in the paper so I'm not suggesting any changes now, but may have some implications on later comments below and would encourage them to explore these as they continue their optimizations in the future.**

In this model, boundary conditions are included into matrix elements, which are stored as an array KT(6,13), where the first dimension corresponds to 6 triangles composing Finite Element, and the second dimension corresponds to 13 types of "wet" points: 1 inside the domain and 12 types of boundary points. This approach is similar to multiplicative masking, as B.C.s are applied by the product to KT, and to unstructured mesh models, as matrix elements are precomputed. The difference from the unstructured mesh models is that only unique elements of the matrix are stored and neighboring points are referenced directly. So, the mask of wet points serves only to restrict the number of computations. We thank the referee and will think in the future how to organize calculations more efficiently.

In our opinion, for the model we have for now, it is not reasonable to change loops' order, as it harms model infrastructure and our parallelization approach, but array indices may be chosen more optimally, setting "depth" index as the first. Nevertheless, this interchange is not crucial for the goals we address in the paper.

**Response:** without changes.

2. **First, the authors show speedups in figure 4 with significantly super-linear speedups in the 2-d case. They attribute this to cache performance without additional evidence (eg from hardware counters or other performance tools). That may be the case, but I think this super-linearity is large enough to warrant further exploration into the cause.**

This super-linear speedup is measured for the code section of approximate length 1000 code lines which consist of 6 loops like in algorithm 3. We stress that these loops have slightly different organization of the calculation and may accelerate in slightly different rates. Some of the loops work with 4D arrays, where the first additional dimension corresponds to "antidiffusive fluxes". Some loops have additional if-conditions, which are needed to perform flux correction in quasi-monotone scheme. Superlinearity occurs at the low-to-middle number of cores, and these cases are usually omitted when scaling up to many cores is shown. Moreover, usually speedups are shown including exchanges, and for this option our speedups are not superlinear.

Finally, from the practical viewpoint, the presented parallelization approach together with the chosen loops/indices ordering may lead to superlinear acceleration. As an example, consider very simple "heat equation" loop:

```fortran
do j = js, je
    do i = is, ie
        do k = 1,depth(i,j)
            Tn(i,j,k) = 0.25_8 * (T(i+1,j,k) + T(i-1,j,k) + T(i,j+1,k) + T(i,j-1,k))
        end do
    end do
end do
```

It accelerates superlinearly at the low-to-middle number of cores for appropriate weights even when MPI exchanges are taken into account (green line in the right subfigure):

[Figure]

Regardless of the actual reason it happens (decrease in the number of cache misses, non-optimal organization of the calculations, or something else), this "heat equation" loop constitutes what we actually intended to do, and there is nothing to optimize here.

**Response:** without changes (see also response to editor).

3. **Third, the large variation in work load at high core counts (fig 5,6) also seems higher than one might expect. As you get fewer points/blocks per core, there will naturally be a little higher variability, but this seems larger than expected and might point to additional problems.**

   As the referee pointed out, balancing could be better. In experiment presented in the manuscript, balancing is limited by the outlier point (figs. 5 and 6), corresponding to the CPU core with 5 blocks and maximum load. This outlier point limits LI to 46%. We have checked balancing optimization procedure (algorithm 4) and found that it doesn't guarantee monotone decrease of LI, as subroutine "remove_not_connected_subdomain" can increase LI. After choosing the best iteration, LI was decreased to 29%. As this behavior is crucial only for "hilbert3d 993 cores" experiment, in the revised manuscript we will update it. Additionally, we have tested METIS multilevel k-way contiguous partitioning algorithm and found that it doesn't give better balancing (LI=39%).

   **Response:** "hilbert3d 993 cores" experiment, which is presented in table 3 and figures 4-9, was updated.

4. **Second, the computational time as a function of wet points seems a bit counter-intuitive (Fig. 5). The authors have shown percentage of wet points rather than total wet points to emphasize their diagnosis again of memory access. But without also seeing the total number of points (computational load), it's a little hard to get a more complete picture. Again, this effect seems too big to attribute solely to cache effects and it seems like more might be going on here.**
   Figures 5 and 6 are provided to assess separately data structure efficiency and load balancing efficiency and clearly show limitations of the described model. As the referee pointed out, the lack of point-to-point correspondence between 5 and 6 figures lead to incomplete picture of what is going on. Here we provide scatter plot (6 figure y axis – 5 figure y axis):

[Figure]

[Figure]

Scatterplots are provided with mean values (solid lines) and maximum values (dashed lines). These values completely define Load Imbalance (LI) in partition and advect3d runtime. As follows from the left figure, spread in runtime is more then spread in the number of computational points. This means that computations are limited by the organization of the calculations, but not by the accuracy of the partitioning algorithm. As advect3d is a function with approximate length of 2500 code lines, which consists of 6 loops like in algorithm 2, and each loop has slightly different organization of the calculations, we claim that overestimation of runtime LI only by 15% in comparison to partition LI is a very good result. In the right figure, there is strict correlation between the number of computational points and advect3d runtime, and computations are limited by the balancing procedure. As this figure is more informative than figure 6, in the revised manuscript we will attach the new figure.

**Response:** Figure 6 is changed to the new one. Also, we update discussion of this figure (lines 247-254 in the revised manuscript):

"Figure 6 additionally shows that spread in runtime cannot be explained by the difference in the number of computational points, i.e. partitioning algorithm works well for 149 cores. Although organization of the calculations may slightly limit model efficiency on a moderate number of cores, it does not limit the model efficiency on 993 cores, where major part of the advect3D runtime spread is explained by the imperfect balancing (see figure 6), but not the data structure (see figure 5). Stagnation of the balancing procedure is evident from the fact that the minimum number of blocks located on a CPU core is 1 for 993 cores, see table 3. Note that computational subdomain corresponding to one CPU core is small enough: on average, it has 9x9 horizontal points with 12 vertical levels for 993 cores."

5. **Minor comments.**
   1) The journal editor will probably mention this, but most references should be changed so that the parentheses are around both author and date unless an integral part of sentence. So for example p1L22-23, should have (FEMAO; Iakolev, 1996, 2012) and (Chernov, 2013; Chernov et al. 2018). And so on throughout the manuscript.
   **Response:** All references are checked.
   2) Fig 2 has cropped the bottom of figures
   **Response:** Fig 2 is shown as we expected. We do not provide axis labels as they correspond to the mesh points, but not to geographical coordinates.
   3) P9L170-180; In this bulleted list, move the text "These three bullets. . ." and "This reduces. . ." after the bulleted list as "The first three bullets. . ." and "The final bullet. . ." Mixing these comments in with the bulleted list was confusing.
   **Response:** done.

[revised manuscript text omitted]